# Differences in the Visual Performances of Patients with Strabismus, Amblyopia, and Healthy Controls

**DOI:** 10.3390/bioengineering9110626

**Published:** 2022-10-29

**Authors:** Danjela Ibrahimi, Juvenal Rodríguez-Reséndiz, Jorge Domingo Mendiola-Santibañez

**Affiliations:** 1Facultad de Medicina, Universidad Autónoma de Querétaro, Querétaro 76176, Mexico; 2Facultad de Ingeniería, Universidad Autónoma de Querétaro, Querétaro 76010, Mexico

**Keywords:** visual performance, strabismus, amblyopia, healthy controls, stereopsis, eye

## Abstract

The aim of this paper is to use the new definition of amblyopia and to define the overall visual performance of healthy controls (HCs), patients with strabismus (PS), and patients with refractive amblyopia (PRA), based on the interaction of selected visual abilities. Method: A total of 398 participants were divided in three groups: HCs, PRA, and PS. Variables such as visual acuity, refractive state, degree of stereopsis, phoria state, magnitude, and type of deviation were analyzed using parametric and non-parametric tests. Results: Binocular visual acuity at near is the unique predictor factor for stereopsis in PRA and PS, while age relates to the amount of binocular visual acuity at near, only for PS with stereopsis. Binocular visual acuity at near and phoria states relate to each other in PRA. Binocular visual acuity at near and far in PS is better than PRA, with no differences in the degree of stereopsis. Stereoblind patients were only found among PS (36%). Only (44.9%) of PS had amblyopia. Exophoria predominated among PRA (69.72%) and HCs (78.87%), while exotropia was the predominant deviation in PS (60.54%). Hyperopia was the predominate refractive error among the groups, HCs (74.65%), PRA (79.82%), and PS (59.85%), followed by astigmatism. Interpretation: HCs perform better than PS and PRA. The visual performance of PS with stereopsis and PRA is similar. Binocular visual acuity at near can predict the degree of stereopsis, and stereoblind patients are exclusively related to strabismus.

## 1. Introduction

Strabismus is a visual disorder, the prevalence of which varies widely in the world [1]. Commonly, strabismus is defined as a misalignment of the visual axis that prevents the retinal fusion of both images and breaks the binocularity of the visual system [2]. Generally, strabismus is followed by other cortico-visual adaptations such as anomalous retinal adaptations, suppression, and amblyopia [3]. Amblyopia is characterized by the loss of visual acuity in one or both eyes, and reflects an abnormal development of the visual cortex [4]. Amblyopia can be the outcome of an imbalanced refractive state as well, in the same way where strabismus can arise as a consequence of amblyopia [5]. No matter how we see it, there is a relationship between amblyopia, strabismus, and the refractive state. Both visual conditions are considered as neuro-developmental disorders, considering that changes in the brain activity and in cortical coherence have been found [6]. Frequently, patients with strabismus and amblyopia present a variety of sensory deficits such as a loss of visual acuity, contrast sensitivity, and a diminishment or lack of stereopsis, which refers to our ability to perceive the three-dimensional space where we live in [7,8]. The sensorimotor imbalance that accompanies strabismus and amblyopia reflects the neural impairment that occurs when normal visual development is disrupted and its remnant on the brain functionality [9]. Neuroscientific research has confirmed alterations in the properties of neurons from the striate and extrastriate cortex [10]. Nonetheless, cortical deficits are not only the properties of V1 and V2, but they make their way to more complex neural pathways and deeper cortical areas [11,12] by affecting many other aspects of visual cognition [13,14]. Strabismus and amblyopia have been associated with difficulties in visual perceptual and visual motor integration skills [15], and the presence of slow brain-waves such as theta-wave, which persists even after a specific treatment [16], illustrates that the brains of patients with strabismus and amblyopia obey their own rules. Considering that 50% of the cortex is related to the visual system, and that more than 30 different cortical areas and 300 intracortical pathways are involved in processing visual information [17], defining the efficacy of the visual system of these patients based on the interaction among selected variables considered crucial for its best functionality, is the purpose of this paper. More specifically, in our research, 398 participants were divided into three groups: healthy controls (HCs), patients with strabismus (PS), and patients with refractive amblyopia (PRA). The scope of this research was to define the overall visual efficacy of our groups based on the interrelationship among our variables of interest. This interaction guided us towards which ones could predict the visual performance of our participants. Motor and sensory visual deficits in PS and PRA were analyzed and compared to HCs. Stereopsis, visual acuity far and near, phoria state, magnitude and type of deviation, refractive state, age, and gender were our variables of interest. A special interest was given to the analysis of the visual efficacy at near, considering visual problems associated with near work experienced during the past few decades [18]. Likewise, it is worth mentioning that our research took place during a pandemic year (2020–2021), and our results could reflect the visual efficacy of participants after experimenting an increase on working and studying at near distances. Previous studies of the visual deficits in patients with strabismus, amblyopia, and anisometropia [7,19,20] have reported sensory deficits in all groups, with one common pattern; decreased or a total loss of binocularity. However, we took this research a step further by making the variables interact so that they could clarify what type of relationship could be established among them, by using the new definition of amblyopia [21]. Over the years, the definition of amblyopia and the proposed acuity criteria for amblyopia have been changing. Since 1942, Feldman and Taylor proposed the <20/50 acuity level. Flom, in 1991, changed it to 20/40, Griffin, in 1982, proposed 20/30, Winter, in 1973, suggested 20/25, and Ciuffreda et al., in 1991, proposed a visual acuity criterion of <20/20. Nowadays, considering the research on the spatial and temporal deficit processing of these patients, amblyopia has been redefined as “a relative reduction in visual acuity greater than one line between the two eyes or an absolute reduction below 20/25 with either eye with associated decrements in visual processing” [21]. This definition allows us to explore and define the visual performance of patients with strabismus and amblyopia when compared to healthy controls, as it includes all those patients with good visual acuity but bad visual efficacy.

## 2. Participants

A total of 398 patients participated in this study (mean age, 9.09 ± 5.05), of which there were 170 females (42.7%; mean age, 10.04 ± 6.02 years) and 228 males (57.3%; mean age, 9.05 ± 4.09 years). Based on the medical histories of participants, and the ophthalmologic and optometric evaluations, they were divided in three groups: patients with strabismus (PS), patients with refractive amblyopia (PRA), and healthy controls (HCs).

### 2.1. Inclusion Criteria

The inclusion criteria were the same for all participants: (i) no history of eye disease (pathologies), (ii) no history of any neurological condition (attention-deficit/hyperactivity disorder, epilepsy, dyslexia), nor psychiatric disease, (iii) no use of medications that could alter the central nervous system (CNS), and (iv) no premature birth.

HCs: (i) best-corrected visual acuity (VA) ≤ 0.1 logMAR units or (=20/25).

PS: (i) Diagnosis of primary strabismus, (ii) no history of previous treatments (surgery or visual training), and (iii) no dissociated or consecutive strabismus.

PRA: (i) Patients with refractive amblyopia, and (ii) best-corrected visual acuity (VA) = 0.2 logMAR units or (≤20/30).

Data on the medical histories of the patients and results from their clinical examinations were collected at the Autonomous University of Querétaro, México, from December 2020 to December 2021. The binocular questionnaires of PS and PRA patients confirmed difficulties and discomforts with their visual performance and processing. The protocols were approved by our Institution’s IRB with approval number 10848, and they conform to the principles of the Declaration of Helsinki. Written informed consent was obtained from the participants or their parents before their enrollment in the study. Eligibility was established after the detailed ophthalmologic and optometric evaluation. All patients had an IQ score in the norm for their chronological age, as confirmed by their medical histories.

### 2.2. Ophthalmologic and Optometric Evaluation

First step: Detailed medical histories regarding the state of their visual health were collected. Based on the medical history, patients who did not meet the inclusion criteria were rejected.

Second step: The motor and sensorial behaviors of the visual system were examined. Motor evaluation is necessary to determine the phoria state, type, and magnitude of strabismus and the ocular movements of the participants. The sensorial evaluation determines the amount of stereopsis, visual acuity at both distances, and the ocular dominance of the participants. Visual acuity was measured with the Bailey-Lovie chart (logMAR) at 3 m and 40 cm, respectively. Stereopsis was measured with the Lang test, which measures three levels of disparities: 1200, 600, and 550 arcminutes, and the Random Dot 2 Test, which goes from gross to fine stereopsis (500 to 12.5 arcminutes).

Third step: Cycloplegic objective refraction using two drops of 1% tropicamide and subjective refraction for the best optical correction was performed. Reevaluation of the sensorimotor behavior of the visual system was performed to observe the impact of the new prescription on the visual system.

Fourth step: Based on the results of the clinical examination, patients were divided into three groups (HCs, PRA, and PS). To achieve better results, ophthalmologic and optometric evaluations were performed during the morning hours. The protocol of the clinical examination can be found described in detail in our previous paper [15].

Figure 1 illustrates the steps followed in this research.

## 3. Statistical Analysis

For the statistical analysis, parametric and non-parametric tests were used. A one way ANOVA with Tukey’s post hoc test for multiple comparison was used to analyze means between three or more groups when N > 30, while Kruskal–Wallis with Bonferroni post hoc analysis was used when N < 30 for each group. The independent T-sample test was used to compare means between two independent groups when N > 30, and Mann–Whitney when N < 30. Multiple regression and correlation analysis were also used to analyze the relationship between one dependent variable and several independent variables, while Pearson correlation was performed to establish the relationship between two variables.

## 4. Results

A total of 398 patients participated in this study, aged 4.11 to 29.11, and divided in three groups: patients with strabismus (PS), refractive amblyopia (PRA), and healthy controls (HCs).

HCs: There were 142 patients in total, 90 male (63.4%) and 52 female (36.6%); the mean age was 9.72 ± 4.74. Of the patients, 19 of them presented with esophoria (13.38%) and 112 had exophoria (78.87%).

Only 13 patients were myopic (9.15%), while 106 were hyperopic (74.65%) (with or without astigmatism); 22 patients did not present any refractive error (15.49%) and only one patient had pure astigmatism (0.71%). Here, 122 patients (85.9%) had right eye motor ocular dominance.

PRA: There were 109 patients in total, 58 male (53.21%) and 51 female (46.79%); the mean age was 9.13 ± 5.26. Here, 15 of them had esophoria (13.76%) and 76 presented with exophoria (69.72%).

Additionally, 17 patients had myopia (15.6%) and 87 presented with hyperopia (79.82%) (with or without astigmatism); five were pure astigmatisms (4.58%). There were 94 patients (86.2%) with right eye motor ocular dominance. All patients presented with stereopsis.

PS: There were 147 patients in total, 80 male (54.42%) and 67 female (45.58%); the mean age was 10.61 ± 6.36. Here, there were 52 patients with ET (35.37%), 89 with XT (60.54%), and six with pure HT (4.09%). Additionally, hypertropia was seen in 16 other patients as a secondary deviation (mean 5.56 ± 4.8 at far and 5.58 ± 5.34 at near, respectively). A total of 53 patients were stereoblind (36%), 94 presented with stereopsis (64%), and 66 patients had amblyopia (44.9%). A total of 111 patients (75.5%) had right eye motor ocular dominance. Furthermore, 27 patients had myopia (18.37%) and 88 presented with hyperopia (59.86%). A total of 24 patients did not present any significative refractive error (16.33%), and eight patients had pure astigmatism (5.44%). It is important to mention that the mean value for PS with stereopsis was (125.81 ± 175.83, N = 94). The mean values and SD of all variables measured in this study are presented in Table 1.

One way ANOVA with Tukey’s post hoc test for multiple comparison was used to analyze most of the variables presented in Table 1, as (N > 30) for each group.

Kruskal–Wallis, on the other hand, was used to analyze the state of myopia.

An independent samples *t*-test (N > 30) was used to analyze the exophoria state at near between PRA and HCs.

The Mann–Whitney test was used to analyze the phoria state between PRA and HCs when N < 30, such as exophoria at far and esophoria (far and near, respectively).

The results were gathered in two categories:1Statistically significant differences among groups

Statistics and results for stereopsis

One way ANOVA was used to analyze differences on the degree of stereopsis among the three groups, where differences were statistically significant: F = 25.64 and *p* < 0.001.

More specifically, *p* < 0.001 when HCs were compared to PA and PS, as seen in Figure 2.

Differences between PRA and PS were statistically insignificant (*p* = 0.25). For PS, stereoblind patients were excluded from the analysis. Additionally, as reflected by the Std, the degree of stereopsis fluctuates widely within subjects in PS, which reflects a deeper imbalance in the sensory system than in PRA.

Additionally, for a second step, emphasis was given on PS and PRA. Patients who did not perceive stereopsis through the Random Test 2 were excluded. A total of 88 PS (mean 90.07 ± 92.79) and 109 PRA (101.61 ± 96.83) had stereopsis with the Random-Dot 2 test. The independent sample *t*-test was used to compare differences on the amount of stereopsis between groups, with no statistically significant difference between them; t = −0.85 and *p* = 0.39. To conclude, the degree of stereopsis measured with Random Dot 2 in PS and PRA is similar.

Results obtained after the visual acuity analysis

The next variable to be analyzed was visual acuity. The results of the one way ANOVA for the visual acuity of the right and left eye at far and near among the groups were as follows:

When the HCs were compared to PRA and PS, for OD, F = 92.71 at far, and F = 110.38 at near, with *p* < 0.001, as seen in Figure 3a,b. For OS, F = 91.93 at far, and F = 102.85 at near, with *p* < 0.001, as seen in Figure 3c,d. However, when PRA and PS were compared, at far, *p* = 0.18 (OD) and *p* = 0.35 (OS) while at near, *p* = 0.03 (OD) and *p* = 0.16 (OS), as seen in Figure 3e. To summarize, HCs had a better visual acuity for both eyes at both distances, whereas statistically significant differences between PRA and PS were only found on the amount of visual acuity of the right eye at near distance.

An independent Sample *t*-Test was used to compare the means of binocular visual acuity at far (BVAF) and binocular visual acuity at near (BVAN) between PRA and PS. Statistically significant differences were found at far (t = −4.31 and *p* < 0.001) and near (t = −2.2 with *p* = 0.03), showing that PS had better binocular visual acuity (BVA) at both distances, as seen in Figure 4a,b. Once more, the mean and Std values for PS indicate the heterogeneity of the values of BVAF and BVAN.

Statistical analysis of the refractive states of participants

A one way ANOVA with Tukey’s post hoc test was then used to analyze the refractive state (hyperopia and astigmatism) between our groups. There were statistically significant differences among the groups. For a hyperopia of OD, F = 31.46, *p* < 0.001, whereas for OS, F = 24.79, *p* < 0.001. Specifically, *p* < 0.001 when HCs are compared to PRA and PS for both eyes, whereas when PRA and PS are compared between them for OD, *p* = 0.63; for OS, *p* = 0.93.

When it comes to the astigmatism value, statistically significant differences among groups were found for OD, F = 4.94, *p* = 0.008, whereas for OS, F = 3.49, *p* = 0.03. More specifically, for OD, differences were found between HCs vs. PS, *p* = 0.01; and HCs vs. PRA, *p* = 0.02. When PRA was compared to PS, the analysis showed a *p* = 0.9. For OS, HCs vs. PRA, *p* = 0.03; HCs vs. PS, *p* = 0.14; PA vs. PS, *p* = 0.67. As it can be seen, the differences are more related to the right eye, being this the dominant one for most of our participants.

The Kruskal–Wallis and the Bonferroni post hoc tests for multiple comparison were used to analyze differences in the mean value of myopia, considering that N < 30 for each group, with no statistically significant differences between the groups. Specifically, H = 0.02, *p* = 0.9 for OD, whereas H = 0.43, *p* = 0.51 for OS.

Multiple regression and correlation analysis for stereopsis and binocular visual acuity

Multiple regression and correlation analysis was also used to analyze the relationship between stereopsis and the amount of binocular visual acuity at far and near for PS and PRA. Binocular visual acuity was considered for this analysis, as the stereopsis test is performed in a binocular state. Table 2 illustrates our findings.

For PRA, a positive correlation was found between stereopsis and the binocular visual acuity (BVA) (R2 = 0.57, F (2,106) = 74.95 *p* < 0.001). However, BVAF was not a unique predictor (β = 208.84, *p* = 0.06), but BVAN was a unique one (β = 999.28, *p* < 0.001), as shown at Figure 5a. Then, for PS, the relationship between BVA (far and near) and stereopsis (only patients with a certain degree of stereopsis were included) was analyzed. A positive correlation was found in this case, R2 = 0.15, F (2,91) = 9.49, *p* < 0.001. However, visual acuity at near was the unique predictor, (β = 278.82, *p* = 0.04), while visual acuity at far was not such a predictor (β = 197.64, *p* = 0.14), as seen in Figure 5b. Considering that there was only one patient with a gross stereopsis of 1200″, it was excluded from the analysis and R2 = 0.19, F (2,90) = 12.1, *p* < 0.001. Once again, binocular visual acuity at near was the unique predictor for stereopsis, β = 251.4, t = 2.52, *p* = 0.01, as seen in Figure 5(b1). Additionally, only PS with amblyopia and stereopsis and PRA were gathered, and the relationship between stereopsis and BVA was analyzed. The patient with gross stereopsis of 1200″ was excluded. There was a positive correlation between amblyopia and stereopsis, R2 = 0.31, F (2,144) = 65.60, *p* < 0.001. Once again, visual acuity at near was the unique predictor, (β = 737.48, *p* < 0.001). BVAF was not such a predictor (β = 244.58, *p* = 0.08), as seen in Figure 5c. To conclude, only visual acuity at near is a predictor for the degree of stereopsis.

For PRA, another step was taken, and the relationship between the amount of BVAN and the amount of exophoria at near was analyzed. The results showed a positive correlation (R2 = 0.09, F (1,71) = 7.06 and *p* = 0.01), as seen in Figure 5d. There is a weak correlation in this case, where β = 16.03, t = 2.66 and *p* = 0.01. Patients with EF were excluded, considering that only a small percentage of our population had EF. Likewise, exophoria at far was not considered for this analysis, as only a few patients presented with this condition.

Statistical analysis of the visual acuity based on age groups

Additionally, the amount of visual acuity at far and near was analyzed based on age.

When PRA and PS with amblyopia were gathered (N = 197), the one way ANOVA analysis showed no statistically differences between groups, F = 1.23/0.34 and *p* = 0.3/0.8 at far and near, respectively. Then, PS and PRA were analyzed separately.

For PRA, the Kruskal–Wallis test showed no significant changes between age groups, with H = 0.29 and *p* = 0.96 at far, while H = 3.46 and *p* = 0.33 at near.

For PS (N = 147), a one way ANOVA analysis showed insignificant statistical differences between age groups; F = 2.5 and *p* = 0.06 at far, while F = 1.17 and *p* = 0.32 at near.

For PS with stereopsis (N = 94), the Kruskal–Wallis test showed statistically significant differences between groups only at near, where H = 9.22 and *p* = 0.03, while at far, H = 5.23 and *p* = 0.16. Refer to Table 3 for more details. A negative relationship was found when the first group was compared to the other three.

In detail, when the visual acuity at near distance was analyzed for PS and stereopsis, the Mann–Whitney test (N < 30 for each group) showed the below results:

Z = −2.65 and *p* = 0.008 when the first and second groups were compared.

Z = −2.25 and *p* = 0.02 when the first and third groups were compared.

Z = −1.95 and *p* = 0.05 when the first and fourth groups were compared. 

Additionally:

Z = −0.65 and *p* = 0.52 for second and third.

Z = −0.19 and *p* = 0.85 for second and fourth.

Z = −0.61 and *p* = 0.54 for third and fourth. 

To summarize these results, it can be concluded that for PS with stereopsis, BVAN obtains lower values in younger children, as seen in Figure 6.

2Statistically insignificant but clinically important findings:

A one way ANOVA showed no age differences between groups, F = 2.35 and *p* = 0.1.

An independent T-sample test showed no age differences between male and female participants, t = −1.51 and *p* = 0.13.

Visual acuity, phoria, and strabismic state analysis

In PS, the magnitude of deviation (ET and XT) was correlated with the amount of visual acuity at far and near, respectively. For patients with ET, multiple regression analysis showed no such correlation; F = 0.19, *p* = 0.83 (both distances). Same results were seen for patients with XT, F = 0.37, *p* = 0.69 at far, and F = 0.1, *p* = 0.9 at near, respectively. Summarizing, visual acuity is not related to the phoria state or the magnitude of deviation.

Stereopsis and gender statistical analysis

An independent samples *t*-test was used to analyze the degree of stereopsis based on gender for each group, with no statistically significant difference between male and female (HCs, t = 0.16, *p* = 0.87; PRA, t = 0.21, *p* = 0.98; PS, t = −0.72, *p* = 0.47). To summarize, gender does not relate to the degree of stereopsis.

Relationship between the refractive state and stereopsis

On another level, the relationship between the amount of hyperopia and stereopsis was analyzed. Considering that only a small percentage of our population was myopic, they were excluded from this analysis. From a total of 398 participants, 281 were hyperopic (70.6%) and only 57 presented myopia (14.3%). Multiple regression and correlation analysis showed no relationship between hyperopia and stereopsis for HCs (F = 0.11, *p* = 0.89), PS (F = 2.7, *p* = 0.07), and PRA (F = 1.13, *p* = 0.33). The amount of hyperopia does not affect the degree of stereopsis.

Refractive state, phoria, and strabismus correlation analysis

The relationship between hyperopia and exophoria at near PRA and HCs, and the magnitude/type of deviation (ET and XT) were now analyzed. Considering that few patients presented exophoria at far for HCs and PRA, they were excluded from this analysis. For HCs, hyperopia was not related to the amount of exophoria at near (F = 0.91, *p* = 0.4), while for PRA, F = 0.09, *p* = 0.92. For PS, F = 0.68, *p* = 0.51 for hyperopia-ET at near distance; F = 1.57, *p* = 0.22 for hyperopia-XT at near; F = 1.78, *p* = 0.18 for hyperopia-ET at far and F = 0.83, *p* = 0.44 for hyperopia-XT at far distance. As revealed by the statistical analysis, the amount of hyperopia does not affect the phoria state or the magnitude of deviation.

Statistical analysis of the amount of stereopsis based on age groups

Finally, for PRA and PS (separately), the degree of stereopsis was analyzed based on age. Four groups were created; 4.11 to 6.11 (first group); 7.0 to 9.11 (second group); 10.0 to 15.0 (third group) and 15.0 to 30.0 (fourth group). The Kruskal–Wallis test was used to compare the means between groups (N = 30 only for the 4.11 to 6.11 group. N < 30 for the rest of the groups). For PRA, H = 5.72 and *p* = 0.13. For PS patients with stereopsis (N = 94), H = 5.99 and *p* = 0.11. Then, PRA (N = 109) and PS with stereopsis (N = 94) were grouped all together. Considering that N > 30 for each group, a one way ANOVA with Tukey’s post hoc was used in this case, with F = 0.85 and *p* = 0.47. Age does not predict the degree of stereopsis.

The relationship between the amount of hyperopia and age groups was analyzed. No relationship was found between age and the refractive state when all patients were analyzed all together (F = 1.1, *p* = 0.29 for hyperopia OD; F = 2.54, *p* = 0.11 hyperopia OS). Refractive state does not depend on age.

The phoria states of PRA and HCs were compared. The amount of exophoria at near was compared using an independent samples *t*-test (N > 30), where t = −1.45 and *p* = 0.15.

A Mann–Whitney Test was used to analyze the phoria state between PRA and HCs when N < 30. Such was the case for exophoria at far and esophoria at far/near, respectively.

For exophoria at far, Z = −0.36 and *p* = 0.72.

For esophoria at far, Z = −0.45 and *p* = 0.65.

For esophoria at near, Z = −0.65 and *p* = 0.51. 

The differences between the phoria states of PRA and HCs were statistically insignificant.

The degree of stereopsis was further analyzed based on the amount of exophoria at near for HCs and PRA, and the magnitude of deviation (ET and XT) for PS. Considering that very few patients presented esophoria at near, they were excluded from this analysis. Likewise, only the phoria state at near was considered, as the stereopsis test is performed at 40 cm. Pearson correlation and multiple regression analysis was used in SPSS. The Pearson correlation showed no relationship between the amount of exophoria at near and the degree of stereopsis: HCs (*p* = 0.16) and PRA (*p* = 0.84). For PS, multiple regression analysis for the far and near tropia states showed (F = 1.84, *p* = 0.17 for XT and F = 1.43, *p* = 0.27 for ET). To conclude, the amount of stereopsis was not related to the phoria state or the magnitude/type of deviation of a patient.

Understanding stereoblind patients:

To understand the characteristics of stereoblind patients, the distribution of frequencies of the magnitudes and types of strabismus were analyzed. They were 53 stereoblind patients, 35 with ET (far and/or near), and 17 XT (far and/or near). Additionally, six patients had hypertropia as a secondary deviation, and only one had pure HT. Figure 6a–d illustrates the results of such a distribution. As seen from Figure 7a,b, 35 patients with ET were stereoblind, which means that ET patients have a greater tendency of being stereoblind. On the contrary, only 17 patients with XT were stereoblind, and a greater amount of deviation is necessary for patients with exotropia to be stereoblind (Figure 7c,d). HT was not graphed as being a secondary deviation.

The distribution of the amount of visual acuity for each eye at both distances for our stereoblind population was graphically presented as well, as seen by Figure 8a–d, where even strabismic patients without amblyopia (VA of 0.0 and 0.1 logMar) can be stereoblind, which confirms that the amount of visual acuity does not predict the presence or absence of stereopsis.

For the same group, a Pearson correlation was used to analyze the relationship between the amount of hyperopia and visual acuity at far/near, where the results were statistically insignificant. For OD, *p* = 0.08/0.1 while for OI, *p* = 0.83/0.6, respectively.

The frequency distribution of the amount of hyperopia for each eye was graphed as well, as illustrated by Figure 9a,b, where even patients with a low degree of hyperopia can be stereoblind, as confirmed by the following graphs.

To conclude, in this research, a thorough statistical analysis was performed to see differences in the visual performances among HCs, PRA, and PS, based on the interactions among selected variables related to their best functionalities. Table 4 represents the statistically significant differences found among them.

## 5. Discussion

During the past few decades, neuroimage and clinical studies have made essential progress on understanding the underlying causes of visual conditions such as amblyopia and strabismus [22,23], with amblyopia being an ideal model for understanding brain plasticity and its critical periods for the recovery of function [22]. As already mentioned, there is a connection between strabismus, amblyopia, and the refractive state (such as anisometropia), and for years, researchers have been focusing on finding differences between strabismic and anisometropic amblyopes [3,8,20]. However, until today, the distinctive patterns of visual loss among groups remain unclear, without any consistent functional pattern [7]. Taking into consideration that the visual system becomes the dominant sensory modality once the child starts to explore the world around him, understanding its development and deficits becomes essential. In this paper, the new definition of amblyopia was considered [21], as it includes all these patients with acceptable visual acuity but with symptoms of discomfort during visual performance.

In this paper, a total of 398 patients participated, divided in three groups: HCs, PRA, and PS. The visual states of patients at far and near were analyzed, and data were compared within subjects and among groups. Visual acuity, stereopsis, phoria state, magnitude, and type of strabismus, age, and gender were our variables of interests. Special interest was given to the visual performance at near, considering the visual problems associated with near work [18].

A thorough statistical analysis was carried out for the purpose of this research. Our goal was to give live to our variables, make them interact, and find the predominate ones that can influence the visual efficacies of the participants. Therefore, the relationship between phoria state and stereopsis, magnitude/type of deviation and stereopsis, visual acuity and stereopsis, refractive state and stereopsis, age, gender and stereopsis, phoria and refractive state, and age and refractive state were explored. In the era of digital work and high technology development, the visual system becomes the main system, and its performance in different aspects of the everyday depend on its efficacity. In this paper, by analyzing the interaction among the chosen variables, we defined which ones are related to each other and we can impact on the visual performances of participants using the new definition of amblyopia for PS and PRA. This research allows us to break an old paradigm and create a new one.

The most interesting finding of this research is that binocular visual acuity at near is the unique predictor of the degree of stereopsis, in both groups, PS and PRA. This result is coherent with the way in which the degree of stereopsis with Random Dot-2 test is measured; at 40 cm and with both eyes opened. In the field of optometry, stereoacuity is an important variable that is associated with the visual performances of our patients, moreso than monocular visual acuity itself [24,25]. Our results confirm the theory that binocular visual acuity at near should be an important variable to be considered for the best achieved stereoacuity of a patient as one of the most essential visual skills [26,27], and the prescription of refractive errors should take into consideration the visual performance at near [28], as visual acuity at far does not relate to stereoacuity. What should be highlighted here is that most visual health professionals prescribe based on the amount of visual acuity at far. As a result, patients with good visual acuity at far present discomfort at near working distance. Based on the results obtained through this research, the old way of evaluating the functionality of the visual system should be reconsidered to make the best of it.

Additionally, the heterogeneity among values obtained by PS patients confirms what the literature highlights, which is that patients with strabismus adopt to their condition as they can, making it difficult to find a common pattern among them [6,7,15], a reason for why analyzing and trying to understand their visual abilities is a challenge for health professionals who are linked to the diagnosis and treatment processes of these patients. However, what makes a difference here is that stereoblind patients were only found among patients with strabismus, whereas patients with refractive amblyopia maintained a certain degree of stereopsis. Additionally, our results are consistent with what other authors have shown, which is that the type and magnitude of deviation, and the amount of visual acuity could not predict the presence or absence of stereopsis [2].

However, what makes a difference here is that stereoblind patients were only found among patients with strabismus, whereas patients with refractive amblyopia maintained a certain degree of stereopsis. Additionally, the type and magnitude of deviation, and the amount of visual acuity could not predict the presence or absence of stereopsis, as reported in previous studies [2].

When it comes to age groups, the only important finding of the statistical analysis was the correlation found for patients with strabismus and stereopsis. Age division was based on previous studies [29,30], and showed that younger children had worse visual acuity at near than older ones. It is known that the 0.0 logMar visual acuity response is not always expected in a patient younger than age 6 [21], which could explain why younger patients do not perform the same as older ones. However, this is not the case for HCs and PRA patients, which reflects differences in the way in which the brain of PS becomes organized to adapt to the motor and sensory difficulties that are associated with strabismus.

The phoria state at near and BVAN in PRA are related to each other. This means that they can affect one another. The importance of this finding is that the amount of visual acuity and the degree of phoria at near should be considered when a patient is treated in visual therapy. Focusing on improving these two variables would improve the overall patients’ visual performance [27].

An expected finding was the excellent visual performance of HCs when compared to PS and PRA. Specifically, they had better visual acuity for each eye at both distances, a higher degree of stereopsis, and less refractive error, without differences in the phoria state when compared to PRA, considering that an exo/esophoria is commonly found in children [31].

In this paper, a consistent functional pattern between PS and PRA was established. BVA at near affects the degree of stereopsis, and PS had better visual acuity at both distances when compared to PRA, with more heterogeneity among the obtained values as they make personal adaptations based on their necessities to compensate for the sensorimotor imbalance of the visual system. However, we could not establish a relationship between the amount of visual acuity at far and stereoacuity [2]. Despite all this, PS and PRA behave differently to HCs, which allow us to visualize that there is an important impact on the functionality of the visual system because of strabismus and amblyopia. We highlight here that the new definition of amblyopia was used in this research, which encompasses a larger population affected by this visual condition, a reason for why our results differ at some points from other studies. In previous papers [7,8,15], the most classic and old definition of amblyopia was considered (≤20/40). However, this definition cannot explain why patients with better visual acuities than 20/40 experience a considerable number of difficulties during visual performance. Additionally, it only represents the visual acuity, as a static concept, but not the whole concept of vision, as a dynamic brain process [21,25,32].

Therefore, using the new definition of amblyopia allows us to overcome these limitations and provide new data on the functionality of the visual system of patients with strabismus and amblyopia. In this paper, we focused on the interaction between selected variables and how they affect the visual performance of the participants, which makes this paper different, as literature offers few information on the interrelationship between visual skills [25]. Likewise, for the first time, the impact of the change in the specific concept of amblyopia on the visual performance was analyzed, which open paths to more research on this topic and create new paradigms in the field of visual health care.

## 6. Conclusions

A total of 398 participants divided into three groups participated in this research: HCs, PRA, and PS. The interaction among selected variables, such as the visual acuity, refractive state, degree of stereopsis, phoria state, magnitude, and type of deviation, were analyzed using parametric and non-parametric tests. As expected, HCs performed better than PRA and PS, while the PRA and PS visual efficacies were similar. Nonetheless, a higher degree of heterogeneity for the measured values was seen in PS. A positive correlation between the BVAN and amount of stereopsis was found in PRA and PS, whereas age for PS with stereopsis was negatively related to the amount of BVAN. Likewise, PS had better BVA at far and near than PRA, but differences in the degree of stereopsis were insignificant. On the other hand, all in the PRA group presented a certain degree of stereopsis (from gross to fine), while 36% of PS were stereoblind. Additionally, almost half of PS had amblyopia (44.9%). In stereoblind patients, the frequency distribution of the type and magnitude of deviation, and visual acuity and refractive error demonstrated the lack of a specific pattern between the state of being stereoblind and these variables. Exophoria predominated among PRA (69.72%) and HCs (78.87%), while exotropia was the predominant deviation in PS (60.54%). In addition, a positive correlation was determined between binocular visual acuity and the phoria state in PRA. Hyperopia was the predominant refractive error among the groups, presented as follows: HCs (74.65%), PRA (79.82%), and PS (59.85%), followed by astigmatism. Gender, refractive error, phoria state, magnitude, and type of deviation did not relate to the stereopsis. No relationship was found between the refractive error and the phoria states or the magnitudes of deviation of the participants. Likewise, visual acuity was not related to the phoria state or the magnitude or type of deviation. No differences were seen in the phoria states of PRA and HCs. To conclude, the overall visual performance of HCs is better when compared to PS and PRA. Interestingly, in the presence of stereopsis, the visual efficacy of PS and PRA is similar. BVAN is the only predictor of the degree of stereopsis for PS and PRA, and stereoblind patients are exclusively related to strabismus.

## Figures and Tables

**Figure 1 bioengineering-09-00626-f001:**
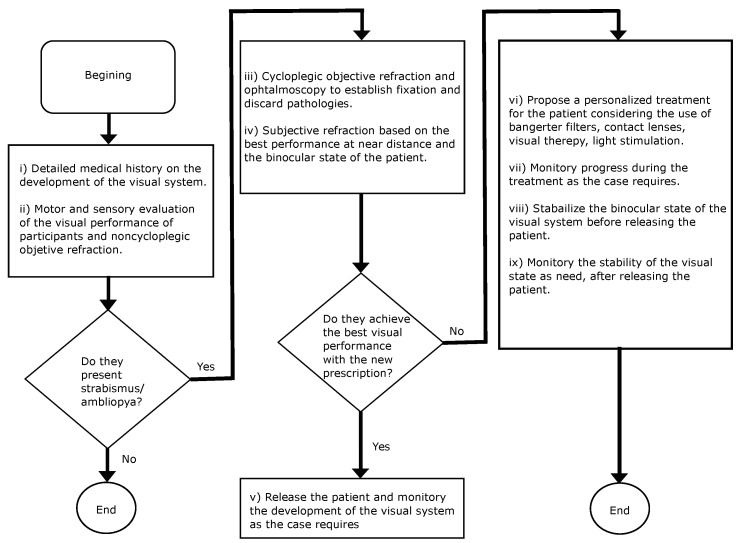
Flowchart in assessing patients, and possible treatments for individuals diagnosed with strabismus and amblyopia.

**Figure 2 bioengineering-09-00626-f002:**
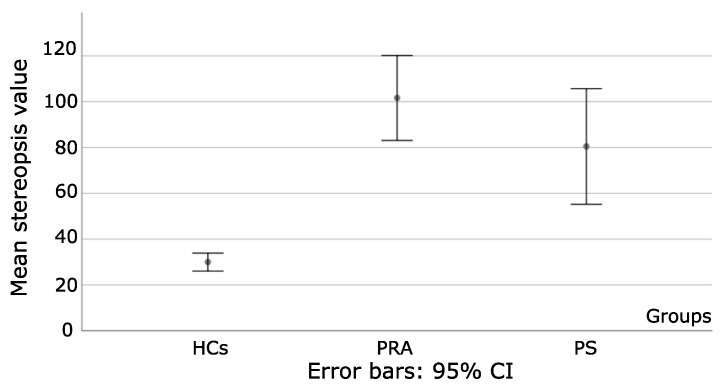
Illustrates the mean values of stereopsis and s.d for each group. PS and PRA present similar degrees of stereopsis (gross stereopsis), but with greater heterogeneity for the obtained values for PS, as shown by the s.d. HCs, which show fine stereopsis, and that allow us to visualize the significant differences between healthy controls and patients with strabismus and amblyopia.

**Figure 3 bioengineering-09-00626-f003:**
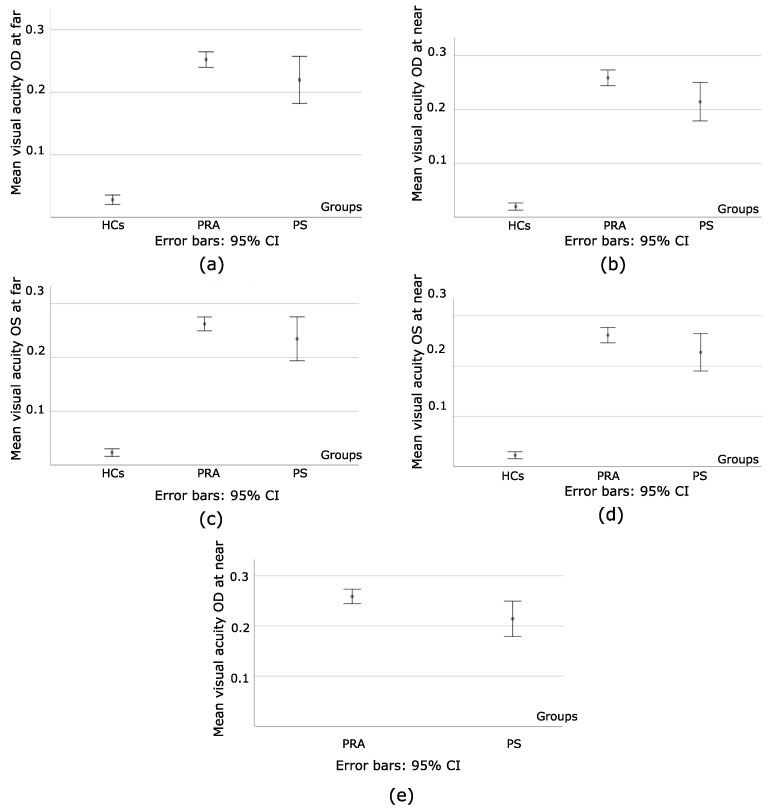
(**a**) Illustrates the mean value of the visual acuity of right eye (OD) at far, and s.d for each group; (**b**) the mean value of the visual acuity of right eye (OD) at near and s.d for each group; (**c**) the mean values of the visual acuity of left eye (OS) at far, and s.d for each group; (**d**) the mean value of the visual acuity of left eye (OS) at near, and s.d for each group; (**e**) the mean value of the visual acuity of right eye (OD) at near and s.d in PRA and PS. What stands out from (**a**–**e**) is the heterogeneity on the obtained values for PS when compared to HCs and PRA. Using HCs as a reference, the impacts of strabismus and amblyopia on the amount of visual acuity of each eye are made evident and more prominent on PS. Nonetheless, PS has better visual acuity than PRA; greater heterogeneities of the obtained values are observed as a result of the sensory and motor deficits that they present.

**Figure 4 bioengineering-09-00626-f004:**
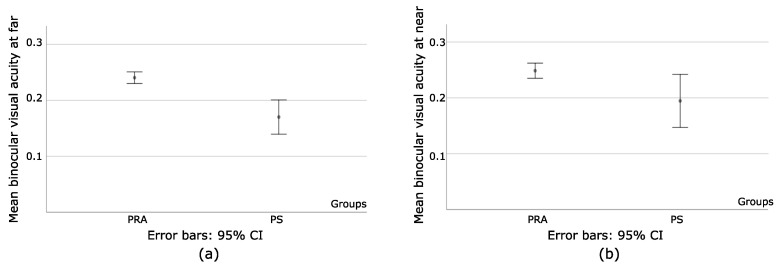
(**a**) Represents the mean value of binocular visual acuity at far and s.d in PRA and PS; (**b**) the mean value of binocular visual acuity at near and s.d in PRA and PS. Despite higher values being obtained in PS, the heterogeneity shown by the s.d reflects a deeper imbalance for the sensory and motor systems at both distances.

**Figure 5 bioengineering-09-00626-f005:**
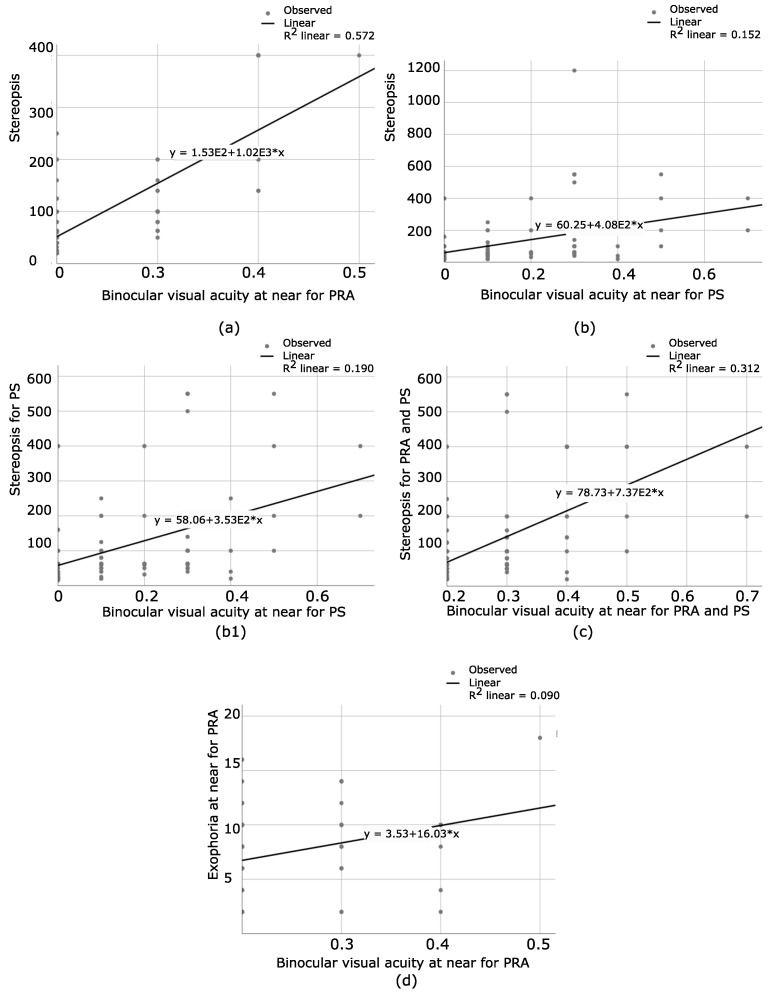
(**a**) Represents the relationship between the degree of stereopsis and the amount of binocular visual acuity at near for PRA. As the visual acuity at near decreases, the degree of stereopsis diminishes. (**b**) The relationship between the degree of stereopsis and the amount of binocular visual acuity at near for PS. Worse stereopsis is obtained in patients with lower visual acuity at near distance. (**b1**) The relationship between the degree of stereopsis and the amount of binocular visual acuity at near for PS, after excluding the only patients with gross stereopsis. R2 increases, which means that the linear relationship becomes stronger. As visual acuity at near decreases, stereopsis deteriorates. (**c**) The relationship between the degree of stereopsis and the amount of binocular visual acuity at near when PRA and PS with amblyopia and stereopsis were put together. There is an important linear relationship, where patients with worse visual acuity at near obtain lower values of stereopsis. (**d**) Shows the relationship between the amount of binocular visual acuity at near and the exophoria state in PRA. A higher degree of exophoria is obtained in patients with lower visual acuity at near, which clarifies that the phoria state deteriorates as the visual acuity at near diminishes.

**Figure 6 bioengineering-09-00626-f006:**
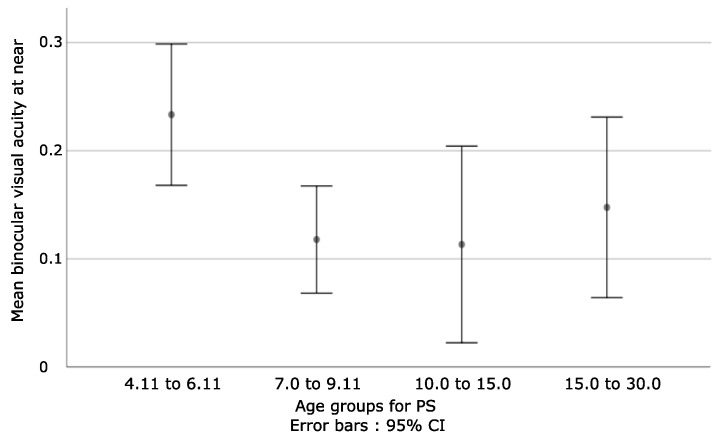
Represents the mean binocular visual acuity at near, defined by age groups for PS with stereopsis. What stands out is that younger children obtain lower values when compared to older ones. However, more heterogeneities for the obtained values are observed in the third and fourth groups, which are probably related to the time spent to make adaptations.

**Figure 7 bioengineering-09-00626-f007:**
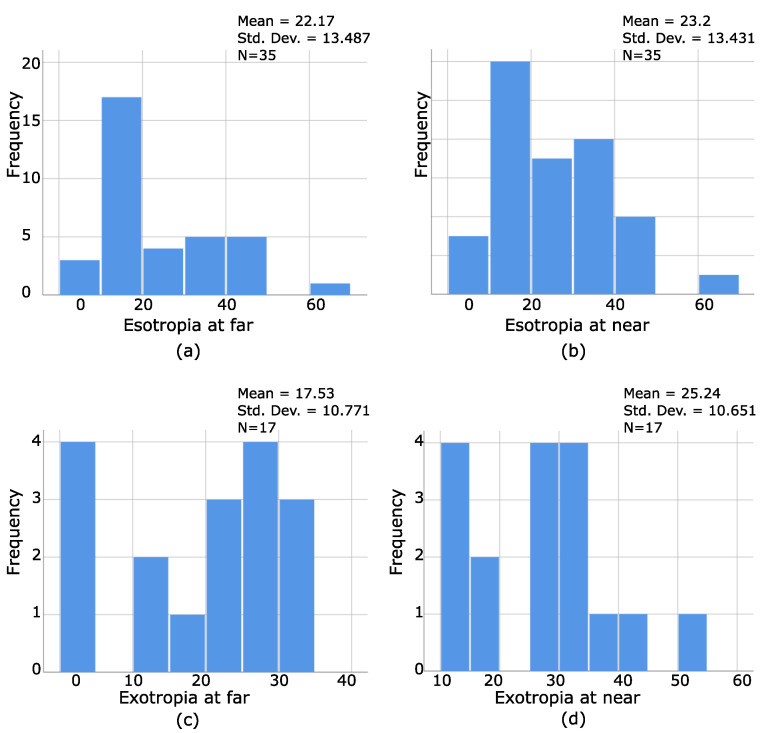
(**a**–**d**) Represents the distribution of frequencies of type and magnitude of deviation in stereoblind PS. As seen by (**a**,**b**), patients with ET have a higher probability of being stereoblind. Additionally, a greater amount of deviation is necessary for patients with exotropia to be stereoblind (**c**,**d**).

**Figure 8 bioengineering-09-00626-f008:**
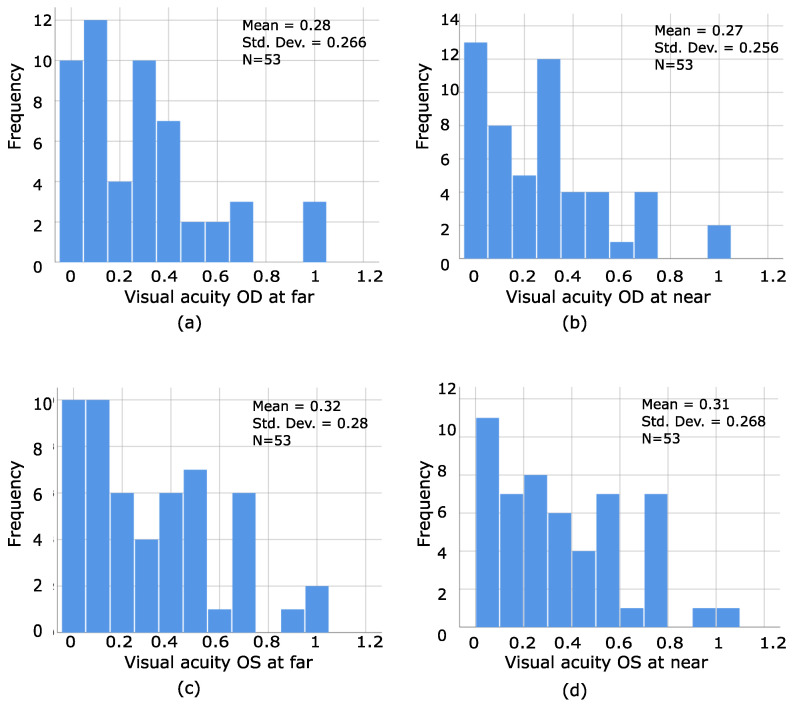
(**a**–**d**) Presents the distribution of frequencies of the amount of visual acuity of the right and left eye at far and near in stereoblind PS. As can be seen, even strabismic patients without amblyopia can be stereoblind, which reaffirms that the amount of visual acuity does not predict the presence or absence of stereopsis.

**Figure 9 bioengineering-09-00626-f009:**
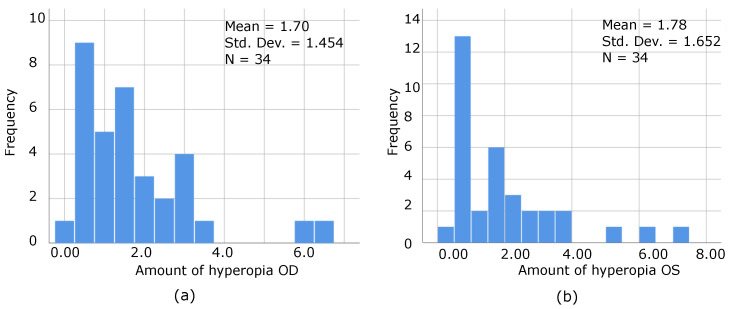
(**a**,**b**) Shows the distribution of frequencies of the amount of hyperopia of the right and left eye in stereoblind PS, where even patients with low degrees of hyperopia can be stereoblind. This reaffirms what statistics have already proven, that the amount of hyperopia does not predict the presence or absence of stereopsis.

**Table 1 bioengineering-09-00626-t001:** Descriptive statistics of the variables analyzed in this study.

	*HCs*	*PRA*	*PS*	*p*-Value
* **Variables** *	**Mean ± Std**	**Mean ± Std**	**Mean ± Std**	
Age	9.72 ± 4.74	9.13 ± 5.26	10.61 ± 6.36	0.1
VA OD F	0.03 ± 0.04	0.25 ± 0.06	0.22 ± 0.23	<0.001
VA OS F	0.02 ± 0.04	0.26 ± 0.07	0.23 ± 0.25	<0.001
VA OD N	0.02 ± 0.04	0.26 ± 0.08	0.21 ± 0.21	<0.001
VA OS N	0.02 ± 0.04	0.26 ± 0.08	0.23 ± 0.23	<0.001
BVAF	0.02 ± 0.04	0.24 ± 0.05	0.17 ± 0.19	<0.001
BVAN	0.02 ± 0.04	0.25 ± 0.07	0.19 ± 0.29	<0.001
Stereopsis	30.0 ± 23.43	101.61 ± 96.83	80.45 ± 152.86	<0.001
EF F/ET F	6.33 ± 6.50	4.0 ± 3.54	19.83 ± 12.41	0.65
EF N/ET N	5.35 ± 3.79	4.62 ± 3.86	20.13 ± 12.43	0.51
XF F/XT F	5.78 ± 4.49	4.94 ± 3.75	9.8 ± 10.36	0.72
XF N/XT N	6.86 ± 3.80	7.74 ± 4.36	17.37 ± 8.71	0.15
Myopia OD	−1.65 ± 1.09	−2.4 ± 1.71	−3.12 ± 2.82	0.9
Myopia OS	−2.00 ± 1.45	−2.54 ± 1.75	−2.6 ± 2.57	0.51
Hyperopia OD	0.75 ± 0.59	1.91 ± 1.05	1.76 ± 1.53	<0.001
Hyperopia OS	0.76 ± 0.58	1.87 ± 1.04	1.81 ± 1.81	<0.001
Astigmatism OD	−1.06 ± 1.17	−1.71 ± 1.13	−1.73 ± 1.42	0.008
Astigmatism OS	−1.11 ± 1.18	−1.68 ± 1.07	−1.51 ± 1.24	0.03

VA, visual acuity; OD, oculus dexter; OS, oculus sinister; BVAF, binocular visual acuity at far; BVAN, binocular visual acuity at near; F, far; N, near; EF, esophoria; XF, exophoria; ET, esotropia; XT, exotropia; Std, standard deviation.

**Table 2 bioengineering-09-00626-t002:** Relationship between the amount of binocular visual acuity at far and near, and the degree of stereopsis for PRA and PS using multiple regression analysis.

*Variables*	*PRA* *N = 108*	*PS* *N = 93*	*PS + PRA* *N = 146*
	t-value *p*-value	t-value *p*-value	t-value *p*-value
VA at far	1.86 0.06	1.5 0.14	1.76 0.08
VA at near	11.66 < 0.001	2.12 0.04	8.1 < 0.001

VA, visual acuity; PRA, patients with refractive amblyopia; PS, patients with strabismus; t-value of independent samples *t*-test.

**Table 3 bioengineering-09-00626-t003:** The relationship between age groups and the amount of visual acuity at near for PS patients with stereopsis (N = 94).

Binocular Visual Acuity at Near						
**Age Groups**	**Mean**	**N**	**Std**	**Age-Group Analysis**	**Z**	* **p** * **-Value**
4.11 to 6.11	0.23	30	0.17			
7.0 to 9.11	0.12	28	0.13	1st vs. 2nd	−2.65	* **0.008** *
10.0 to 15.0	0.11	15	0.16	1st vs. 3rd	−2.25	* **0.02** *
15.0 to 30.0	0.15	21	0.18	1st vs. 4th	−1.95	* **0.05** *
Total	0.16	94	0.17			

The Mann–Whitney test (*n* < 30 for each group) was used to compare means between groups.

**Table 4 bioengineering-09-00626-t004:** A summary of the statistically significant differences for the visual performance among groups.

		Stereo	VA OD (N/F)	VA OS (N/F)	BVAF	BVAN	RS (H)	RS (A)
HCs	PRA	X	X	X	X	X	X	
	PS	X	X	X	X	X	X	X
PRA	PS		X (N)		X	X		
BVAN	PRA	X						
	PS	X						
BVAN	PRA/PS with stereo	X						
Age group	PS with stereo					X		
F	PRA					X		

PS, patients with strabismus; VA OD (N/F), visual acuity oculus dexter at far and near; VA OS (N/F), visual acuity oculus sinister at far and near; RS (H), refractive state for hyperopia; RS (A), refractive state for astigmatism; stereo, stereopsis; BVAF, binocular visual acuity at far; BVAN, binocular visual acuity at near; XF, exophoria.

## Data Availability

The data presented in this study are available on request from the corresponding author. The data are not publicly available due to confidentiality.

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
