# Peer review of "Differences in the Visual Performances of Patients with Strabismus, Amblyopia, and Healthy Controls"

_bioengineering, 2022, doi:10.3390/bioengineering9110626_

Round 1

Reviewer 1 Report

The manuscript entitled "Differences in the Visual Performance of Patients with Strabismus, Amblyopia and Healthy Controls" is based on a new definition of amblyopia assessed in healthy controls, patients with strabismus, and patients with refractive amblyopia. This study provides objective measurements of various parameters of visual abilities and compares them in these cohorts. The topic adds to the current literature and is of clinical interest, especially considering the potential importance of binocular visual acuity at near and stereopsis in these patients, which are normally not evaluated in suspect amblyopic patients.

The paper is well-planned, and the analysis correctly reflects the points expressed in the results and discussion sections. The cohorts are adequately sized to reflect the results. The tables and figures work well to describe the data. The English can use minor revisions for enhanced flow. The findings regarding common patterns of hyperopia as refractive error, followed by astigmatism, right eye dominance, and exotropia in strabismus confirm data in the literature.

I have only one comment regarding therapeutic options in patients with strabismus and amblyopia. The authors should consider including a flowchart in assessing patients and possible treatments for individuals diagnosed with these conditions. This can provide enhanced clinical usefulness of the paper.  

Reviewer 2 Report

This a clinical study with the corresponding statistical analysis of the results. My only concern is that I was expecting a real engineering paper as the name of the journal suggests. But there is a lack of signal processing or machine learning approach to the problem. It is basically a clinical paper better suited for a clinical/opthalmology 100% merdical journal.

Authors obtain some interesting conclusions analyzing the results (some correlated variables), but nothing else is explored (a regression model? a classification model based on your data?

Discussion section seems a mixture of summary/conclusions (in fact it overlaps with conclusion section). You can shorten and focus on the discussion (correlation findings)
